# Medications Modulating the Acid Sphingomyelinase/Ceramide System and 28-Day Mortality among Patients with SARS-CoV-2: An Observational Study

**DOI:** 10.3390/ph16081107

**Published:** 2023-08-04

**Authors:** Nicolas Hoertel, Katayoun Rezaei, Marina Sánchez-Rico, Alfonso Delgado-Álvarez, Johannes Kornhuber, Erich Gulbins, Mark Olfson, Charles Ouazana-Vedrines, Alexander Carpinteiro, Céline Cougoule, Katrin Anne Becker, Jesús M. Alvarado, Frédéric Limosin

**Affiliations:** 1INSERM U1266, Université Paris Cité, F-75014 Paris, France; 2Service de Psychiatrie et Addictologie de l’Adulte et du Sujet Agé, DMU Psychiatrie et Addictologie, Hôpital Corentin-Celton, GHU APHP.Centre, F-92130 Issy-les-Moulineaux, France; 3Department of Psychobiology and Behavioural Sciences Methods, Faculty of Psychology, Universidad Complutense de Madrid, 28223 Madrid, Spain; 4Department of Biological and Health Psychology, Faculty of Psychology, Universidad Autónoma de Madrid, 28049 Madrid, Spain; 5Department of Psychiatry and Psychotherapy, University Hospital, Friedrich-Alexander-University of Erlangen-Nuremberg (FAU), 91054 Erlangen, Germany; 6Institute of Molecular Biology, University Hospital Essen, University of Duisburg-Essen, 47057 Essen, Germanykatrin.becker-flegler@uni-due.de (K.A.B.); 7Department of Psychiatry, New York State Psychiatric Institute, Columbia University, New York, NY 10032, USA; 8Service de Psychiatrie de l’Adulte, DMU Psychiatrie et Addictologie, Hôpital Hôtel-Dieu, AP-HP, Université Paris Cité, F-75004 Paris, France; 9Department of Hematology and Stem Cell Transplantation, University Hospital Essen, University of Duisburg-Essen, 47057 Essen, Germany; 10Institute of Pharmacology and Structural Biology (IPBS), University of Toulouse, CNRS, 31000 Toulouse, France

**Keywords:** COVID-19, SARS-CoV-2, mortality, FIASMA, ceramide, antidepressant

## Abstract

Prior evidence indicates the potential central role of the acid sphingomyelinase (ASM)/ceramide system in the infection of cells with SARS-CoV-2. We conducted a multicenter retrospective observational study including 72,105 adult patients with laboratory-confirmed SARS-CoV-2 infection who were admitted to 36 AP-HP (Assistance Publique–Hôpitaux de Paris) hospitals from 2 May 2020 to 31 August 2022. We examined the association between the ongoing use of medications functionally inhibiting acid sphingomyelinase (FIASMA), which reduces the infection of cells with SARS-CoV-2 in vitro, upon hospital admission with 28-day all-cause mortality in a 1:1 ratio matched analytic sample based on clinical characteristics, disease severity and other medications (N = 9714). The univariate Cox regression model of the matched analytic sample showed that FIASMA medication use at admission was associated with significantly lower risks of 28-day mortality (HR = 0.80; 95% CI = 0.72–0.88; *p* < 0.001). In this multicenter observational study, the use of FIASMA medications was significantly and substantially associated with reduced 28-day mortality among adult patients hospitalized with COVID-19. These findings support the continuation of these medications during the treatment of SARS-CoV-2 infections. Randomized clinical trials (RCTs) are needed to confirm these results, starting with the molecules with the greatest effect size in the study, e.g., fluoxetine, escitalopram, and amlodipine.

## 1. Introduction

The COVID-19 pandemic is still regarded as a leading concern due to its deleterious effects on public health, healthcare infrastructure, and the economy [1,2,3,4,5,6]. There remains an unmet need for effective outpatient treatments for Coronavirus Disease 2019 (COVID-19), particularly for low- and middle-income countries, especially treatments that can be taken orally, have few medical contraindications [7,8], and are well-tolerated, affordable, and readily available [9,10,11,12].

Prior evidence indicates that the ASM/ceramide system may play an important role in the infection of cells with SARS-CoV-2 [13]. Acid sphingomyelinase (ASM) is an enzyme that cleaves sphingomyelin into ceramide, forming gel-like platforms in the plasma membrane. Experimental in vitro studies support the notion that SARS-CoV-2 causes the activation of the acid sphingomyelinase/ceramide pathway, which facilitates viral entry into cells through these gel-like platforms, favoring the clustering of activated SARS-CoV-2 cellular ACE2 receptors [13] (Figure 1). Therefore, it was shown that medications with the functional inhibition of acid sphingomyelinase (FIASMA), which inhibit ASM and reduce the formation of ceramide-enriched membrane platforms [12], decrease cell infection with SARS-CoV-2 and subsequent inflammation [12,13,14,15]. FIASMA medications include certain antidepressants (e.g., fluoxetine, fluvoxamine, escitalopram, amitriptyline), calcium channel blockers (e.g., amlodipine, bepridil), antihistamine medications (e.g., hydroxyzine and promethazine), and other specific medications [16]. In addition, drugs such as fluoxetine have also been shown to act directly on the virus and its replication, respectively. It remains to be determined whether different functional inhibitors of acid sphingomyelinase act on the acid sphingomyelinase/ceramide system and additional targets that are also important for infection, thereby amplifying the effects of the drugs used against the infection.

Evidence from preclinical studies suggests that the infection of Vero E6 cells with SARS-CoV-2 can be hindered through the inhibition of the ASM/ceramide system by specific antidepressants, such as escitalopram, fluoxetine, or ambroxol [13,15,17]. The addition of ceramides to cells treated with these medications restores the infection [13]. In healthy volunteers, the infection of freshly isolated nasal epithelial cells with SARS-CoV-2 was blocked after the oral administration of amitriptyline [13]. Other studies conducted with human and nonhuman host cells confirmed the in vitro antiviral activity of several FIASMA antidepressants against different variants of SARS-CoV-2 [18,19,20,21,22,23,24,25]. Finally, the results from a K18-hACE2 mouse model of SARS-CoV-2 infection support the antiviral and anti-inflammatory properties of fluoxetine, possibly explained by the modulation of the ceramide system [17].

Several clinical trials have strengthened this preclinical evidence. Observational cohort studies of COVID-19 patients have indicated that FIASMA antidepressants and the FIASMAs amlodipine and hydroxyzine are associated with a reduced risk of mechanical ventilation or death in the acute care setting [26,27,28,29,30,31,32] and a decreased risk of hospital or emergency department visits among outpatients [33]. A systematic review and meta-analysis of six randomized controlled trials (RCTs) (N = 4197) found that a medium dose of the FIASMA antidepressant fluvoxamine (100 mg twice a day) was significantly associated with reduced mortality, hospitalization, and hospitalization/emergency department visits and not associated with increased serious adverse events [34]. Finally, two observational, multicenter, retrospective cohort studies conducted at Greater Paris University Hospitals showed that FIASMA medications, mostly FIASMA antidepressants, calcium channel blocker medications, and hydroxyzine, were significantly associated with a decreased likelihood of death or intubation [26,28] among inpatients with COVID-19.

Taken together, these results favor the possible repurposing of FIASMA medications against COVID-19. However, the few prior observational studies explored a limited range of FIASMA molecules (e.g., only FIASMA antidepressants [33] or the FIASMA hydroxyzine [35]), and several of them examined composite outcomes, such as intubation or death [26,28], posing challenges for the interpretation of the results.

In this report, we examined the link between the use of FIASMA medications at hospital admission and 28-day mortality among adult COVID-19 patients hospitalized at 36 Greater Paris University Hospitals. We hypothesized that FIASMA medication use would be associated with diminished mortality among COVID-19 inpatients.

## 2. Results

### 2.1. Characteristics of the Cohort

Of 72,105 adult patients hospitalized with COVID-19, 261 patients (0.4%) were excluded due to missing data (Figure 2).

Of the remaining 71,844 inpatients, 2354 patients (3.3%) were excluded because they took a FIASMA medication after their admission to hospital. Of the remaining 69,490 patients, 4857 (7.0%) received a FIASMA medication at the time of hospital admission, and 64,633 did not. Twenty-eight-day mortality occurred in 4416 (6.8%) patients. The associations of the clinical characteristics with 28-day mortality and the use of FIASMA medications at hospital admission are shown in Appendix A (Table A1 and Table A2). In the matched analytic sample, no covariate substantially differed between groups (all SMDs < 0.1) (Table A3).

### 2.2. Twenty-Eight-Day Mortality

In the matched analytic sample, 28-day mortality occurred in 625 patients (12.9%) who took a FIASMA medication at admission and in 772 patients (15.9%) who did not. The univariate Cox regression model in the matched analytic sample showed a significant association between FIASMA medication use at baseline and a reduced risk of 28-day mortality (HR = 0.80; 95% CI = 0.72–0.88; *p* < 0.001) (Figure 3; Table 1), corresponding to an ARR of death of 2.7% and an NNT of 37. This association remained significant when stratifying by age, sex, and period of hospitalization (Figure 4; Table 1; Table A4).

Exploratory analyses indicated that the use of FIASMA cardiovascular system medications (particularly other FIASMA cardiovascular system medications) and FIASMA nervous system medications (particularly FIASMA psychoanaleptic medications) was significantly associated with reduced 28-day mortality (Table 2; Table A4). For most individual FIASMA molecules, the hazard ratios were lower than 1. For all non-significant associations, the post hoc estimates of statistical power ranged from 3.5% to 59.6% (Table A5). Fluoxetine, amlodipine, and escitalopram were significantly associated with reduced 28-day mortality.

## 3. Discussion

In this multicenter, observational, retrospective study, the use of a FIASMA medication was significantly linked to reduced 28-day mortality, independent of sociodemographic characteristics, psychiatric and other medical comorbidities, COVID-19 severity, or other medications. The magnitude of this association (HR = 0.80; 95% CI = 0.72–0.88; *p* < 0.001) corresponded to an ARR of death of 2.7% and an NNT of 37. This association held in multiple sensitivity analysis. Additional exploratory analyses suggested that FIASMA cardiovascular system medications, particularly amlodipine, and FIASMA nervous system medications, particularly fluoxetine and escitalopram, were significantly associated with decreased 28-day mortality.

These results confirm and extend the preclinical [13,15,16,18,19,20,21,22,23,24,25,36,37], computational molecular docking [38], observational [26,27,28,29,30,31,32,33,34,35,39], and clinical [40,41,42,43,44,45] study findings suggesting that the ASM/ceramide system may play an important role in SARS-CoV-2 infection, particularly in the case of the FIASMA medications fluoxetine [17,46,47], escitalopram [27,29], and amlodipine [32,48]. These findings are also in line with studies indicating that clinical severity and inflammation markers in patients with COVID-19 are significantly associated with sphingomyelinase and ceramidase activity and the plasma levels of ceramides [3,4,5,17,49,50,51].

Th inhibition of the ASM [37,52] by FIASMA medications may result in antiviral effects (through the diminution of ceramide-enriched membrane domains resulting in decreased viral entry and subsequent inflammation) and anti-inflammatory effects (through the inhibition of this enzyme in endothelial and immune cells [9,11,12]). Because fluoxetine had the largest effect size in this study and has one of the strongest in vitro effects on the ASM [52], is well-tolerated [53,54], and is in the World Health Organization’s Model List of Essential Medicines, this molecule should be prioritized for randomized clinical trials in patients with COVID-19 [29].

The protective associations of FIASMA medications may also result from complex interactions between different biological mechanisms. These mechanisms may include anti-inflammatory properties, either through the high affinity of certain FIASMA medications for sigma-1 receptors (S1Rs) (e.g., fluoxetine and fluvoxamine) or through their effects on non–S1R-IRE1 pathways (e.g., nuclear factor κ B, peroxisome proliferator-activated receptor γ, Toll-like receptor 4, or inflammasomes) [47,55,56,57], reduced mast cell degranulation, decreased platelet aggregation, increased melatonin levels, interference with endolysosomal viral trafficking, and antioxidant properties [55,56,57]. The relative contribution of each mechanism may vary depending on disease stage, the dose prescribed, and the delay of treatment initiation.

This study has strengths, including its assessment of numerous potential confounders, such as markers of clinical severity, its substantial sample size, and the large period of observation, making relevant to different SARS-CoV-2 variants.

This study also has limitations. First, observational studies have two potential biases: unmeasured confounding and confounding by indication. Although the analyses were adjusted for numerous potential confounders, such as sex, age, psychiatric and other medical conditions, and markers of COVID-19 severity, it is still possible that some residual confounding remained unmeasured. For example, information on vaccination status and obesity was not available. In addition, we were unable to adjust our analyses for all the 36 AP-HP hospitals and all the medications, including non-FIASMA psychotropic medications, due to concerns regarding collinearity among these variables and the presence of zero events of a contingency table in some cells, including a high number of degrees of freedom. Second, a causal relationship cannot be established based on our observational study, and RCTs are necessary to confirm these results [58]. Third, information on medication discontinuation was not available, which might have contributed to an underestimation of the magnitude of the observed associations. Fourth, information on patients’ nutrition, which may play a significant role in immune system functioning and overall health [59], was not available. Fifth, even though we used a multicenter study design, the results may not be generalizable to other regions or to outpatients [60]. Finally, due to the rapidly evolving nature of the COVID-19 pandemic, including the emergence of new variants, changes in preventive measures, and evolving treatment protocols, future studies would benefit from evaluating whether FIASMA are still active against infections with new virus variants [61].

## 4. Materials and Methods

### 4.1. Setting and Cohort Assembly

We conducted a multicenter retrospective cohort study at 36 AP-HP hospitals from 2 May 2020 to 31 August 2022 [29], including all adults aged 18 years or over who had been hospitalized at these medical centers with COVID-19. COVID-19 was ascertained using a positive reverse transcriptase–polymerase chain reaction (RT-PCR) test of nasopharyngeal or oropharyngeal swab specimens. The sample in this study did not overlap with the samples of the two previous studies focusing on FIASMA medications and using the AP-HP Warehouse data [26,28], which had a different inclusion period (i.e., from 24 February 2020 to 1 May 2020).

This observational study received approval from the Institutional Review Board of the AP-HP Clinical Data Warehouse (decision CSE-20- 20_COVID19, IRB00011591, 8 April 2020) [10,26,27,28,29,35,62,63,64,65,66,67,68]. AP-HP Clinical Data Warehouse initiatives ensure informed patient consent regarding the different studies approved through a transparency portal in accordance with the European Regulation on data protection and authorization, n°1980120, from the National Commission for Information Technology and Civil Liberties (CNIL).

### 4.2. Data Sources

The AP-HP Health Data Warehouse (‘Entrepôt de Données de Santé (EDS)’) contains all available clinical data on all inpatient visits for COVID-19 to 36 Greater Paris University Hospitals. The data included patient demographic characteristics, vital signs, laboratory test and RT-PCR test results, medication administration data during hospitalization, current medical diagnoses, and death certificates.

### 4.3. Variables Assessed

All variables assessed are detailed in Table A1. The sociodemographic characteristics included sex, age, hospital location, hospitalization period, psychiatric and non-psychiatric medical conditions based on the ICD-10 diagnosis codes during the visit, and medications prescribed according to compassionate use or as part of a clinical trial. The dates of medication prescriptions were recorded. Disease clinical and biological severity were also assessed. Clinical severity was defined based on at least one of the four following criteria [69,70]: resting peripheral capillary oxygen saturation in ambient air < 90%, respiratory rate > 24 breaths/min or <12 breaths/min, temperature > 40 °C, or systolic blood pressure < 100 mm Hg. Biological severity was considered to be met if the plasma lactate levels were higher than 2 mmol/L or in the case of a low lymphocyte-to-C-reactive protein ratio or high neutrophil-to-lymphocyte ratio [71] (both severity variables were binarized at the median value in the full sample).

### 4.4. FIASMA Medications

FIASMA medications were defined as medications displaying a residual in vitro ASM activity < 50%, as described in detail elsewhere [11,36]. We classified the medications following their Anatomical Therapeutic Chemical (ATC) codes (as detailed in Table 2).

FIASMA medication use was defined as having a prescription of at least one FIASMA medication at the time of hospital admission and at least one prior prescription of the same molecule within the last 6 months.

### 4.5. Study Baseline and Endpoint

The study baseline was the date of hospital admission. The endpoint was 28-day all-cause mortality. Patients without an endpoint event had their data censored at 28 days of follow-up.

### 4.6. Statistical Analysis

We calculated the frequency of each baseline characteristic described above for the adult inpatients with COVID-19 taking or not taking a FIASMA medication at baseline and compared them using standardized mean differences (SMDs) [72,73,74]. We considered SMDs greater than 0.1 to reflect significant differences [73].

To examine the association between FIASMA medication use at baseline and the risk of mortality during the 28 days following admission, we used Cox proportional hazard regression models [75] in a matched analytic sample of inpatients with COVID-19 receiving or not receiving a FIASMA medication. In order to reduce the effects of confounding variables, we used a 1:1 ratio matched analytic sample based on sex, age, hospital, period of hospitalization, medications prescribed as part of a clinical trial or according to compassionate use, psychiatric and other medical comorbidities, and biological and clinical markers of COVID-19 severity. Specifically, we used the nearest matching method [76]. We performed additional multivariable Cox regression models, including all unbalanced covariates (i.e., with a SMD > 0.1) [73].

If the main association was significant, we planned to calculate both the between-group difference in absolute risk reduction/increase (ARR) and the number needed to treat (NNT), considering a weighted time-to-event design.

To test the robustness of the primary analysis, we performed sensitivity analyses and separately reproduced the above-mentioned analyses (i) in women and men, (ii) in younger and older patients (based on the median age of the fully matched analytic sample), and (iii) in two different periods of hospitalizations (based on the median date of hospitalization in the fully matched analytic sample).

As an exploratory analysis, we reproduced the above-mentioned analyses for each class of FIASMA medications and individual FIASMA molecules. We selected, a priori, one control for each case of exposure to each class of FIASMA medications and five controls for each exposed case of exposure to each individual FIASMA molecule.

We performed residual analyses for all the associations to determine the fit of the data and checked the assumptions, including multicollinearity diagnoses, using the generalized variance inflation factor (GVIF) for all the multivariable analyses. Our proportional hazard assumption was verified using proportional hazard tests and diagnostics based on weighted residuals [75] for all the survival analyses. Finally, we examined the potential presence and influence of outliers. We also performed post hoc statistical power calculations for all the associations, assuming a 20% mortality reduction. All analyses were conducted in R software version 3.6.3 (R Project for Statistical Computing), and statistical significance was fixed a priori at a two-sided *p*-value < 0.05. We followed the recommendations of the Strengthening the Reporting of Observational Studies in Epidemiology (STROBE) Initiative [77].

## 5. Conclusions

In this multicenter, observational, retrospective study, the ongoing use of functional inhibitors of acid sphingomyelinase (FIASMA) medications at hospital admission was significantly and substantially associated with reduced 28-day mortality, independent of sociodemographic characteristics, psychiatric or other medical comorbidities, the severity of the infection, or other medications among adult inpatients with COVID-19. This association held true in multiple sensitivity analyses. Additional exploratory analyses indicated that FIASMA cardiovascular system medications, particularly amlodipine, and FIASMA nervous system medications, particularly fluoxetine and escitalopram, were also associated with reduced 28-day mortality. These findings support the continuation of these medications during the treatment of SARS-CoV-2 infections. Randomized clinical trials (RCTs) against placebos as well as recommended antiviral treatments are needed to confirm these results, starting with fluoxetine, escitalopram, and amlodipine, which displayed the most robust results in our study [17,29,33,34,78].

## Figures and Tables

**Figure 1 pharmaceuticals-16-01107-f001:**
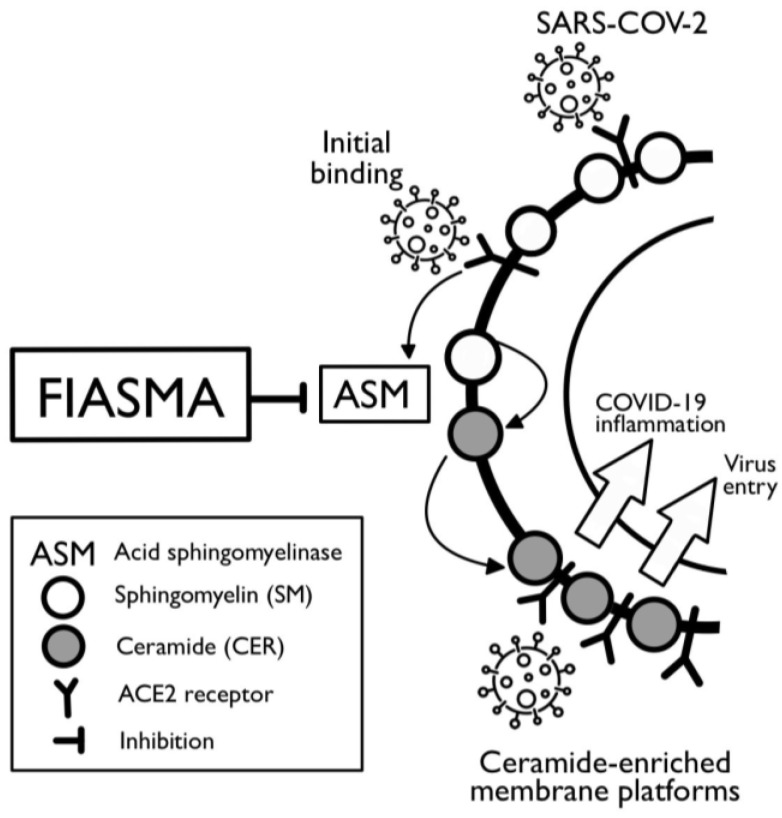
Biological mechanisms proposed by Carpinteiro et al. [13,15], underlying the potential effects of the functional inhibitors of acid sphingomyelinase (FIASMAs) on SARS-CoV-2 infection. SARS-CoV-2 may activate the acid sphingomyelinase/ceramide pathway, which, in turn, facilitates viral entry into cells through gel-like platforms that favor the clustering of activated SARS-CoV-2 cellular ACE2 receptors. Inhibition of the ASM by FIASMAs may result in a reduced concentration of ceramides, decreased viral entry, and subsequent inflammation.

**Figure 2 pharmaceuticals-16-01107-f002:**
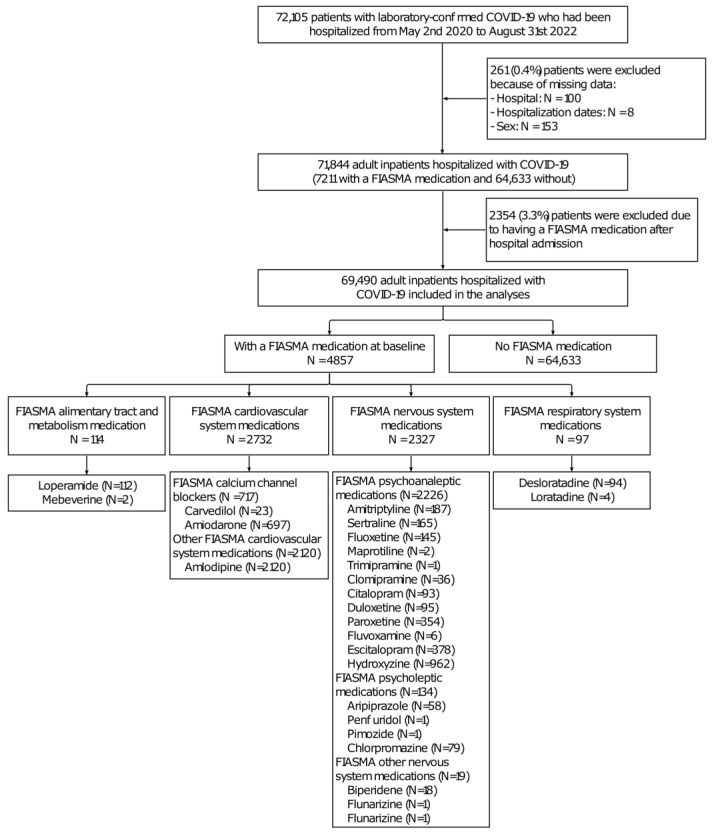
Study cohort.

**Figure 3 pharmaceuticals-16-01107-f003:**
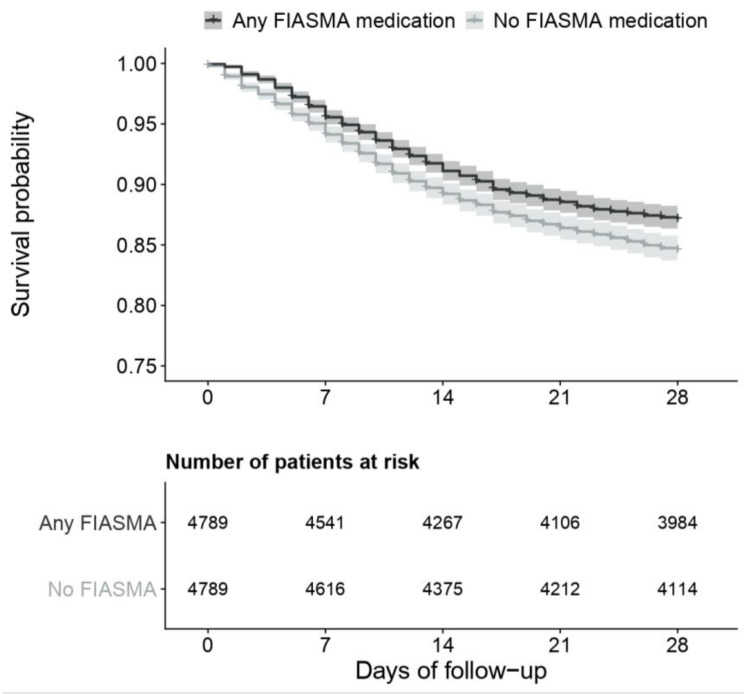
FIASMA medication use and 28-day mortality in the matched analytic sample (N = 9714).

**Figure 4 pharmaceuticals-16-01107-f004:**
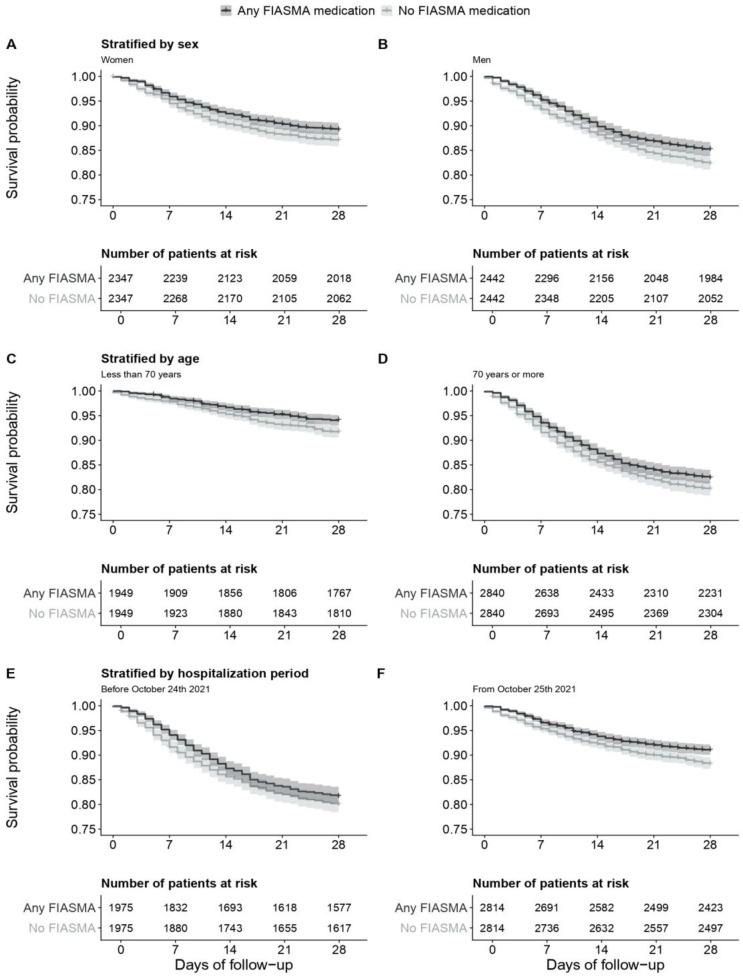
FIASMA medication use and 28-day mortality in the matched analytic sample, stratified by sex (**A**,**B**), age (**C**,**D**), and period of hospitalization (**E**,**F**).

**Table 1 pharmaceuticals-16-01107-t001:** FIASMA medication use at hospital admission and 28-day all-cause mortality in the matched analytic sample of adult inpatients with COVID-19.

	Number of Events/Number of Patients	Crude Cox Regression Analysis of the Matched Analytic Sample
	N/N (%)	HR (95%CI; *p*-Value)
Full sample (N = 9714)		
FIASMA medication	625/4857 (12.9%)	0.80 (0.72–0.88; <0.001)
No FIASMA medication	772/4857 (15.9%)	Ref.
Women (N= 4744)		
FIASMA medication	258/2372 (10.9%)	0.80 (0.68–0.94; 0.007 *)
No FIASMA medication	318/2372 (13.4%)	Ref.
Men (N= 4970)		
FIASMA medication	367/2485 (14.8%)	0.82 (0.71–0.94; 0.004 *)
No FIASMA medication	441/2485 (17.7%)	Ref.
Younger (≤70 years) (N = 3940)		
FIASMA medication	117/1970 (5.9%)	0.70 (0.55–0.88; 0.003 *)
No FIASMA medication	166/1970 (8.4%)	Ref.
Older (>70 years) (N= 5774)		
FIASMA medication	508/2887 (17.6%)	0.84 (0.74–0.94; 0.003 *)
No FIASMA medication	594/2887 (20.6%)	Ref.
Hospitalized before 24 October 2021 (N= 2037)		
FIASMA medication	372/2037 (18.3%)	0.85 (0.74–0.98; 0.021 *)
No FIASMA medication	431/2037 (21.2%)	Ref.
Hospitalized from 25 October 2021 (N= 5640)		
FIASMA medication	253/2820 (9.0%)	0.67 (0.57–0.79; <0.001 *)
No FIASMA medication	368/2820 (13.0%)	Ref.

* Two-sided *p*-value is significant (*p* < 0.05). Abbreviations: HR, hazard ratio; CI, confidence interval; Ref., reference group.

**Table 2 pharmaceuticals-16-01107-t002:** Use of FIASMA medications at hospital admission and 28-day all-cause mortality in the matched analytic samples of adult inpatients with COVID-19.

	Patients with Medication	Patients without Medication in the Matched Sample ^a^	Crude Cox Regression Analysis in the Matched Analytic Sample	Multivariable Cox Regression Analysis of the Matched Analytic Sample Adjusted for Unbalanced Covariates
	N/N (%)	N/N (%)	HR (95%CI; *p*-Value)	AHR (95%CI; *p*-Value)
FIASMA alimentary tract and metabolism medication	13/114 (11.4%)	12/114 (10.5%)	1.10 (0.50–2.41; 0.816)	1.41 (0.61–3.24; 0.420) ^b^
*Loperamide*	13/112 (11.6%)	67/560 (12.0%)	0.98 (0.54–1.77; 0.944)	0.98 (0.54–1.78; 0.953) ^c^
*Mebeverine*	0/2 (0.0%)	1/10 (10.0%)	NA	NA
FIASMA cardiovascular system medications	389/2732 (14.2%)	490/2732 (17.9%)	0.78 (0.68–0.89; <0.001 *)	NP
FIASMA calcium channel blockers	152/717 (21.2%)	157/717 (21.9%)	0.97 (0.77–1.21; 0.774)	NP
*Carvedilol*	3/23 (13.0%)	10/115 (8.7%)	1.50 (0.41–5.46; 0.537)	1.82 (0.48–6.82; 0.377) ^d^
*Amiodarone*	151/697 (21.7%)	711/3485 (20.4%)	1.07 (0.90–1.28; 0.429)	NP
Other FIASMA cardiovascular system medications	256/2120 (12.1%)	368/2120 (17.4%)	0.67 (0.57–0.79; <0.001 *)	0.69 (0.58–0.80; <0.001 *) ^e^
*Amlodipine*	256/2120 (12.1%)	1857/10600 (17.5%)	0.67 (0.59–0.76; <0.001 *)	0.66 (0.58–0.75; <0.001 *) ^f^
FIASMA nervous system medications	266/2327 (11.4%)	332/2327 (14.3%)	0.79 (0.67–0.92; 0.004 *)	0.83 (0.71–0.98; 0.024 *) ^g^
FIASMA psychoanaleptic medications	256/2226 (11.5%)	310/2226 (13.9%)	0.81 (0.69–0.96; 0.014 *)	0.93 (0.79–1.10; 0.382) ^h^
*Amitriptyline*	28/187 (15.0%)	131/935 (14.0%)	1.06 (0.71–1.60; 0.772)	1.24 (0.82–1.87; 0.306) ^i^
*Sertraline*	21/165 (12.7%)	138/825 (16.7%)	0.75 (0.47–1.19; 0.218)	0.82 (0.52–1.30; 0.395) ^j^
*Fluoxetine*	9/145 (6.2%)	100/725 (13.8%)	0.44 (0.22–0.87; 0.019 *)	0.49 (0.25–0.97; 0.042 *) *^k^*
*Maprotiline*	0/2 (0.0%)	0/10 (0.0%)	NA	NA
*Trimipramine*	0/1 (0.0%)	1/5 (20.0%)	NA	NA
*Clomipramine*	7/36 (19.4%)	21/180 (11.7%)	1.73 (0.74–4.07; 0.209)	2.07 (0.86–5.00; 0.104) ^l^
*Citalopram*	18/93 (19.4%)	69/465 (14.8%)	1.35 (0.8–2.27; 0.254)	1.42 (0.83–2.41; 0.197) ^m^
*Duloxetine*	7/95 (7.4%)	54/475 (11.4%)	0.65 (0.30–1.44; 0.291)	0.78 (0.35–1.74; 0.548) ^n^
*Paroxetine*	45/354 (12.7%)	253/1770 (14.3%)	0.88 (0.64–1.21; 0.420)	0.88 (0.64–1.20; 0.417) ^o^
*Fluvoxamine*	0/6 (0.0%)	4/30 (13.3%)	NA	NA
*Escitalopram*	45/378 (11.9%)	323/1890 (17.1%)	0.67 (0.49–0.91; 0.012 *)	0.69 (0.51–0.95; 0.022 *) ^p^
*Hydroxyzine*	104/962 (10.8%)	591/4810 (12.3%)	0.88 (0.71–1.08; 0.210)	1.09 (0.89–1.35; 0.396) ^q^
FIASMA psycholeptic medications	10/134 (7.5%)	11/134 (8.2%)	0.91 (0.39–2.14; 0.824)	1.04 (0.43–2.50; 0.936) ^r^
*Aripiprazole*	1/58 (1.7%)	13/290 (4.5%)	NA	NA
*Penfluridol*	0/1 (0.0%)	0/5 (0.0%)	NA	NA
*Pimozide*	0/1 (0.0%)	0/5 (0.0%)	NA	NA
*Chlorpromazine*	9/79 (11.4%)	29/395 (7.3%)	1.57 (0.74–3.32; 0.237)	1.87 (0.87–4.00; 0.107) ^s^
Other FIASMA nervous system medications	3/19 (15.8%)’	4/19 (21.1%)	NA	NA
*Biperidene*	3/18 (16.7%)	13/90 (14.4%)	NA	NA
*Flunarizine*	0/1 (0.0%)	0/5 (0.0%)	NA	NA
FIASMA respiratory system medications	11/97 (11.3%)	13/97 (13.4%)	0.83 (0.37–1.86; 0.654)	1.61 (0.66–3.91; 0.297) ^t^
*Desloratadine*	11/94 (11.7%)	62/470 (13.2%)	0.88 (0.46–1.67; 0.700)	0.92 (0.49–1.76; 0.807) ^u^
*Loratadine*	0/4 (0.0%)	1/20 (5.0%)	NA	NA

^a^ The ratio was set a priori at 1:1 for categories of molecules and at 1:5 for individual molecules. ^b^ Adjusted for age, hospital, period of hospitalization, any respiratory disorder, any disease of the musculoskeletal system, diseases of the genitourinary system, any eye–ear–nose–throat disorder, biological severity of COVID-19 at baseline, and clinical severity of COVID-19 at baseline. ^c^ Adjusted for age and any diseases of the genitourinary system. ^d^ Adjusted for age, sex, period of hospitalization, any neoplasm or disease of the blood, any cardiovascular disorder, any respiratory disorder, and biological severity of COVID-19 at baseline. ^e^ Adjusted for age. ^f^ Adjusted for age and any medication according to compassionate use or as part of a clinical trial. ^g^ Adjusted for age. ^h^ Adjusted for age, any respiratory disorder, and biological severity of COVID-19 at baseline. ^i^ Adjusted for age, any cardiovascular disorder, any respiratory disorder, any endocrine disorder, and biological severity of COVID-19 at baseline. ^j^ Adjusted for age, sex, any other infectious disease, and biological severity of COVID-19 at baseline. ^k^ Adjusted for age, sex, hospital, and biological severity of COVID-19 at baseline. ^l^ Adjusted for age, hospital, any medication according to compassionate use or as part of a clinical trial, any mental disorder, and biological severity of COVID-19 at baseline. ^m^ Adjusted for age, hospital, period of hospitalization, medications prescribed as part of a clinical trial or according to compassionate use, neoplasms and diseases of the blood, respiratory disorders, endocrine disorders, and clinical severity of COVID-19 at baseline. ^n^ Adjusted for age, hospital, any respiratory disorder, and biological severity of COVID-19 at baseline. ^o^ Adjusted for hospital. ^p^ Adjusted for age and any respiratory disorder. ^q^ Adjusted for age, any respiratory disorder, any endocrine disorder. ^r^ Adjusted for age and hospital. ^s^ Adjusted for age, any mental disorder, any disease of the musculoskeletal system, any diseases of the genitourinary system, and biological severity of COVID-19 at baseline. ^t^ Adjusted for age, sex, hospital, any medication as part of a clinical trial or according to compassionate use, any other infectious disease, any mental disorder, any respiratory disorder, any digestive disorder, any endocrine disorder, and biological severity of COVID-19 at baseline. ^u^ Adjusted for hospital, period of hospitalization, and biological severity of COVID-19 at baseline. * Two-sided *p*-value is significant (*p* < 0.05). Abbreviations: HR, hazard ratio; CI, confidence interval; NA, not applicable; NP, not performed due to the lack of unbalanced variables.

## Data Availability

Data from the AP–HP Health Data Warehouse can be obtained upon request at https://eds.aphp.fr//. The statistical code is available upon request.

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
