# Peer review of "Medications Modulating the Acid Sphingomyelinase/Ceramide System and 28-Day Mortality among Patients with SARS-CoV-2: An Observational Study"

_pharmaceuticals, 2023, doi:10.3390/ph16081107_

Round 1

Reviewer 1 Report

  1. In the abstract and introduction sections, FIASMA is mentioned without proper elaboration and it is not clear how it relates to "the acid sphingomyelinase/ceramide system". Further clarification is needed.

  2. The method section should provide information on how data heterogeneity was addressed, particularly among different hospitals and different psychoanaleptic medications. This aspect could be further discussed in the discussion section.

  3. It would be beneficial to state and explain any limitations of this observational study in the discussion section, if there are any such limitations. Especially with the continuous mutation of the new coronavirus and the constantly changing situation of epidemic prevention and control, are the recommended treatment methods facing limitations in terms of safety or effectiveness?

no comments in this point.

Author Response

July 12th, 2023,

Prof. Dr. Amélia Pilar Rauter

Editor-in-Chief, Pharmaceuticals

Dear Prof. Rauter,

Thank you for the opportunity to submit a revised version of manuscript R1-2395894, “Medications modulating the acid sphingomyelinase/ceramide system and 28-day mortality among inpatients with SARS-CoV-2: an observational study”. We want to thank the reviewers for their helpful comments, which have allowed us to strengthen this manuscript. A point-by-point response is included below, and the corresponding changes have been made in the manuscript and have been highlighted in yellow in a separate file.

The revised manuscript contains no data, patient information, or other material or results that have been published or are in press or submitted elsewhere.

We look forward to hearing from you and thank you in advance for considering this contribution.

Best Regards,

Nicolas HOERTEL,

On behalf of all the authors

Reviewer 1

1/ In the abstract and introduction sections, FIASMA is mentioned without proper elaboration and it is not clear how it relates to "the acid sphingomyelinase/ceramide system". Further clarification is needed.

Answer: We thank the reviewer for this comment. In the abstract and introduction, we now more clearly elaborate the relationship between FIASMA medications and the ASM/ceramide system, and its hypothesized role in SARS-CoV-2 infection.  

Abstract: “Prior evidence indicates the potential central role of the acid sphingomyelinase (ASM)/ceramide system in the infection of cells by SARS-CoV-2.”

Abstract: “We examined the association of ongoing use at hospital admission of medications functionally inhibiting the acid sphingomyelinase (FIASMA), which reduce the infection of cells by SARS-CoV-2 in vitro, with 28-day all-cause mortality, in a 1:1 ratio matched analytic sample based on clinical characteristics, disease severity, and other medications (N=9,714).”

P2, L58-67: “Prior evidence indicates that the ASM/ceramide system may play an important role in the infection of cells by SARS-CoV-2 [13]. Acid sphingomyelinase (ASM) is an enzyme that cleaves sphingomyelin into ceramide, forming gel-like platforms in the plasma membrane. Experimental in vitro studies support that SARS-CoV-2 causes activation of the acid sphingomyelinase/ceramide pathway, which facilitates viral entry into the cells through these gel-like platforms favoring the clustering of activated SARS-CoV-2 cellular receptors ACE2 [13] (Figure 1). Therefore, it was shown that medications with functional inhibition of acid sphingomyelinase (FIASMA), which inhibit ASM and reduce the formation of ceramide-enriched membrane platforms [12], decrease cell infection with SARS-CoV-2 and subsequent inflammation [12–15].”

2/ The method section should provide information on how data heterogeneity was addressed, particularly among different hospitals and different psychoanalytic medications. This aspect could be further discussed in the discussion section.

Answer: To address data heterogeneity and reduce the effects of confounding, we performed Cox proportional hazards regression models in 1:1 ratio matched analytic samples based on a wide range of clinical characteristics, including sex, age, hospital, period of hospitalization, medications prescribed as part of a clinical trial or according to compassionate use, psychiatric and other medical comorbidities (i.e., other infectious diseases, mental disorders, neoplasms and diseases of the blood, diseases of the nervous system, cardiovascular disorders, respiratory disorders, digestive disorders, dermatological disorders, diseases of the musculoskeletal system, diseases of the genitourinary system, endocrine disorders, and eye-ear-nose-throat disorders), and biological and clinical markers of COVID-19 severity. Specifically, we used the nearest matching method. We performed additional multivariable Cox regression models including all unbalanced covariates (i.e., with SMD>0.1).

We tried to reduce indication bias by taking into account a large number of psychiatric and other medical diagnoses in the analyses. We were unable to adjust analyses for all the 36 AP-HP hospitals and all medications, including non-FIASMA psychotropic medications, due to collinearity among these variables and the presence of zero events in some cells of a contingency table with a large number of degrees of freedom.

Following this comment, we have now better explained how we addressed data heterogeneity in the methods, and added this limitation:

P12, L301-307: “In order to reduce the effects of confounding, we used a 1:1 ratio matched analytic sample based on sex, age, hospital, period of hospitalization, medications prescribed as part of a clinical trial or according to compassionate use, psychiatric and other medical comorbidities, and biological and clinical markers of COVID-19 severity. Specifically, we used the nearest matching method [76]. We performed additional multivariable Cox regression models including all unbalanced covariate (i.e., with SMD>0.1) [73].”

P9, L226-235: “This study also has limitations. First, observational studies have two potential biases: unmeasured confounding and confounding by indication. Although the analyses were adjusted for numerous potential confounders, such as sex, age, psychiatric and other medical conditions, and markers of COVID-19 severity, it is still possible that some residual confounding remained unmeasured. For example, information on vaccination status and obesity was not available. In addition, we were unable to adjust analyses for all the 36 AP-HP hospitals and all medications, including non-FIASMA psychotropic medications, due to collinearity concern among these variables and the presence of zero events in some cells of a contingency table including a high number of degrees of freedom.”

3/ It would be beneficial to state and explain any limitations of this observational study in the discussion section, if there are any such limitations. Especially with the continuous mutation of the new coronavirus and the constantly changing situation of epidemic prevention and control, are the recommended treatment methods facing limitations in terms of safety or effectiveness?

Answer: Following this comment, we added this important point in the discussion section as follows:

P9-10, L241-245: “Finally, due to the rapidly evolving nature of the COVID-19 pandemic, including the emergence of new variants, changes in preventive measures, and evolving treatment protocols, future studies would benefit in evaluating whether FIASMA are still active against infections with new virus variants [61].”

Reviewer 2

4/ On abstract section Page No. 1 Line No. 32 “inpatients” does you mean “patients” if so, the correct the same.

Answer: We thank the reviewer for noticing this typo, which has now been corrected.

5/ The word “FIASMA” used on Page No. 1 Line No. 34 without abbreviation. The word should be abbreviated first and then can be used in short form throughout the manuscript. However, “RCTs” is also not abbreviated. Check whole manuscript and correct the same.

Answer:  We have carefully rechecked the whole manuscript to ensure that all abbreviations are defined.

6/ “The FIASMA cardiovascular system medications mainly amlodipine, and FIASMA nervous system medications, in particular fluoxetine and escitalopram, were associated with decreased 28-day mortality”. Are there any additional studies that supports this if so, then add reference of it.

Answer: Following this comment, we have added additional studies supporting these findings, as follows:

P9, L197-204: “These results confirm and extend preclinical [13,15,16,18–25,36,37], computational molecular docking [38], observational [26–35,39], and clinical [40–45] study findings suggesting that the ASM/ceramide system could play an important role in SARS-CoV-2 infection, and in particular the FIASMA medications fluoxetine [17,46,47], escitalopram [27,29] and amlodipine [32,48]. These findings are also in line with studies indicating that clinical severity and inflammation markers in patients with COVID-19 are significantly associated with sphingomyelinase and ceramidase activity and plasma levels of ceramides [3–5,17,49–51].”

7/ There are few other drugs such as remedesvir, camostat mesylate, nafamostat etc are showing the best results against the SARS-CoV-2. In this manuscript the other drugs are not compared. If possible, add the importance of “FIASMA” as compared to other drugs. For this you can refer the research paper given bellow.

1) Kailas D Sonawane, Sagar S Barale, Maruti J Dhanavade, Shailesh R Waghmare, Naiem H Nadaf, Subodh A Kamble, Ali Abdulmawjood Mohammed, Asiya M Makandar, Prayagraj M Fandilolu, Ambika S Dound, Nitin M Naik, Vikramsinh B More (2021) Structural insights and inhibition mechanism of TMPRSS2 by experimentally known inhibitors Camostat mesylate, Nafamostat and Bromhexine hydrochloride to control SARS-coronavirus-2: A molecular modeling approach, Informatics in medicine unlocked, 24, 100597.

Answer: We agree with the reviewer that RCTs testing fluoxetine, escitalopram, and amlodipine against placebo as well as against recommended antiviral drugs are needed to confirm our results. We thank the reviewer for suggesting this reference, which is now included in the manuscript. Following this comment, we modified the manuscript as follows:

P13, L341-344: “Randomized clinical trials (RCTs) against placebo as well as against recommended anti-viral treatments are needed to confirm these results, starting with fluoxetine, escitalopram, and amlodipine, which displayed the most robust results in our study [17,29,33,34,78].”

8/ There is strong relation between patients’ nutrition and Covid-19. It will be interesting to note the relation between “FIASMA” and effect of nutrition. The point can be added including which type of nutrition is best for speedy recovery during Covid-19. For this please refer the research paper given below.

1) Dhanavade, Maruti J.; Sonawane, Kailas Dashrath (2022) Role of Nutrition in COVID19: Present Knowledge and Future Guidelines, 18, 6, 516-517.

Answer: We agree with the reviewer and thank her/him for suggesting this reference, which is now included in the manuscript as follows:

P9, L238-240: “Fourth, information on patients’ nutrition, which may play a significant role in immune system functioning and overall health [59], was not available.”

9/ The study showed that “fluoxetine, escitalopram, and amlodipine, which displayed the most remarkable results” why is this so? This statement should be supported by the experimental studies to find the exact mechanism of action. Do these drugs inhibit SARS-CoV-2? Or it only acts on human cells to create difficulties in viral replication? This has to be studied or supported with any reference.

Answer: Following this important comment, we have detailed results from prior experimental work supporting the contributive role of this mechanism of action in SARS-CoV-2 infection and motivating the realization of this observational study. We also have extended the reference list. We modified the manuscript as follows:

P2, 58-75: “Prior evidence indicates that the ASM/ceramide system may play an important role in the infection of cells by SARS-CoV-2 [13]. Acid sphingomyelinase (ASM) is an enzyme that cleaves sphingomyelin into ceramide, forming gel-like platforms in the plasma membrane. Experimental in vitro studies support that SARS-CoV-2 causes activation of the acid sphingomyelinase/ceramide pathway, which facilitates viral entry into the cells through these gel-like platforms favoring the clustering of activated SARS-CoV-2 cellular receptors ACE2 [13] (Figure 1). Therefore, it was shown that medications with functional inhibition of acid sphingomyelinase (FIASMA), which inhibit ASM and reduce the fomation of ceramide-enriched membrane platforms [12], decrease cell infection with SARS-CoV-2 and subsequent inflammation [12–15]. FIASMA medications include certain antidepressants (e.g., fluoxetine, fluvoxamine, escitalopram, amitriptyline), calcium channel blockers (e.g., amlodipine, bepridil), antihistamine medications (e.g., hydroxyzine and promethazine), and other specific medications [16]. In addition, drugs such as fluoxetine have been also shown to act directly on the virus and its replication, respectively. It remains to be determined whether different functional inhibitors of the acid sphingomyelinase act on the acid sphingomyelinase/ceramide system and additional targets that are also important for infection, and thereby amplify the effect of the drugs against infection.”

P3, L83-93: “Evidence from preclinical studies suggest that the infection of Vero E6 cells with SARS-CoV-2 can be hindered by the inhibition of the ASM/ceramide system by specific antidepressants, such as escitalopram or fluoxetine, or ambroxol [13,15,17]. The addition of ceramides in cells treated with these medications restores the infection [13]. In healthy volunteers, the infection of freshly isolated nasal epithelial cells with SARS-CoV-2 was blocked after oral administration of amitriptyline [13]. Other studies conducted with human and nonhuman host cells confirmed in vitro antiviral activity of several FIASMA antidepressants against different variants of SARS-CoV-2 [18–25]. Finally, results form a K18-hACE2 mouse model of SARS-CoV-2 infection support the antiviral and anti-inflammatory properties of fluoxetine, possibly explained by the modulation of the ceramide system [17].”

P9, L205-212: “Inhibition of the ASM [37,52] by FIASMA medications may result in antiviral effects (through the diminution of ceramide-enriched membrane domains resulting in decreased viral entry and subsequent inflammation) and anti-inflammatory effects (through inhibition of this enzyme in endothelial and immune cells [9,11,12]). Because fluoxetine has the largest effect size in this study and one of the strongest in vitro effects on the ASM [52], is well-tolerated [53,54] and on the World Health Organization’s Model List of Essential Medicines, this molecule should be prioritized for randomized clinical trials in patients with COVID-19 [29].”

10/ Is there any computational study available on “FIASMA” drugs? If so, then can be added for better understanding.

Answer:  Following this comment, we have added a reference to the work of Naz et al. (2023) using a computational molecular docking and dynamic simulation approach applied to dock a drug library containing 257 functional inhibitors of ASM, as follows (please also refer to the response to comment #9 above):

P10, L197-204: “These results confirm and extend preclinical [13,15,16,18–25,36,37], computational molecular docking [38], observational [26–35,39], and clinical [40–45] study findings suggesting that the ASM/ceramide system could play an important role in SARS-CoV-2 infection, and in particular the FIASMA medications fluoxetine [17,46,47], escitalopram [27,29] and amlodipine [32,48]. These findings are also in line with studies indicating that clinical severity and inflammation markers in patients with COVID-19 are significantly associated with sphingomyelinase and ceramidase activity and plasma levels of ceramides [3–5,17,49–51].”

11/ A figure or graphical abstract can be added that explains role of “FIASMA” drugs in relation with Covid-19 disease progression and inhibition. This will enhance understanding of reader.

Answer: We thank the reviewer for this comment. Following this suggestion, we have added a Figure explaining the potential role of “FIASMA” medications in relation with Covid-19 disease progression and inhibition (please see Figure 1).

12/ There is not any conclusion. Write the conclusion section with at least 200 words giving all the important findings.

Answer: Following this comment, we have added the following conclusion:

Page 11-12, L332-344: “In this multicenter observational retrospective study, ongoing use of functional inhib-itors of acid sphingomyelinase (FIASMA) medications at hospital admission was signifi-cantly and substantially associated with reduced 28-day mortality, independently of so-ciodemographic characteristics, psychiatric and other medical comorbidity, severity of the infection, and other medications, among adult inpatients with COVID-19. This associa-tion held in multiple sensitivity analyses. Additional exploratory analyses indicated that FIASMA cardiovascular system medications, in particular amlodipine, and FIASMA nervous system medications, in particular fluoxetine and escitalopram, were also associ-ated with reduced 28-day mortality. These findings support continuation of these medica-tions during treatment of SARS-CoV-2 infections. Randomized clinical trials (RCTs) against placebo as well as against recommended antiviral treatments are needed to con-firm these results, starting with fluoxetine, escitalopram, and amlodipine, which dis-played the most robust results in our study [17,29,33,34,78].”

13/ Overall the manuscript is having good data but should be presented with less error and more scientific information. Proofread the whole manuscript and correct all the grammatical mistakes before moving further with decision.

Answer: Following this comment, we carefully reviewed the whole manuscript while correcting all grammatical mistakes and adding precise scientific information.

Reviewer 3

14/ In the introduction write more details about the mechanism of action behind the antidepressants or ambroxol effects

Answer: Following this comment, we have provided more details concerning results from prior experimental work supporting the contribution of this mechanism of action in SARS-CoV-2 infection. We also have also added a Figure explaining the potential role of “FIASMA” medications in relation with Covid-19 disease progression and inhibition (please see Figure 1).

We modified the manuscript as follows:

P2, L58-70: “Prior evidence indicates that the ASM/ceramide system may play an important role in the infection of cells by SARS-CoV-2 [13]. Acid sphingomyelinase (ASM) is an enzyme that cleaves sphingomyelin into ceramide, forming gel-like platforms in the plasma membrane. Experimental in vitro studies support that SARS-CoV-2 causes activation of the acid sphingomyelinase/ceramide pathway, which facilitates viral entry into the cells through these gel-like platforms favoring the clustering of activated SARS-CoV-2 cellular receptors ACE2 [13] (Figure 1). Therefore, it was shown that medications with functional inhibition of acid sphingomyelinase (FIASMA), which inhibit ASM and reduce the for-mation of ceramide-enriched membrane platforms [12], decrease cell infection with SARS-CoV-2 and subsequent inflammation [12–15]. FIASMA medications include certain antidepressants (e.g., fluoxetine, fluvoxamine, escitalopram, amitriptyline), calcium channel blockers (e.g., amlodipine, bepridil), antihistamine medications (e.g., hydroxyzine and promethazine), and other specific medications [16].”

P3, L83-93: “Evidence from preclinical studies suggest that the infection of Vero E6 cells with SARS-CoV-2 can be hindered by the inhibition of the ASM/ceramide system by specific antidepressants, such as escitalopram or fluoxetine, or ambroxol [13,15,17]. The addition of ceramides in cells treated with these medications restores the infection [13]. In healthy volunteers, the infection of freshly isolated nasal epithelial cells with SARS-CoV-2 was blocked after oral administration of amitriptyline [13]. Other studies conducted with human and nonhuman host cells confirmed in vitro antiviral activity of several FIASMA antidepressants against different variants of SARS-CoV-2 [18–25]. Finally, results form a K18-hACE2 mouse model of SARS-CoV-2 infection support the antiviral and anti-inflammatory properties of fluoxetine, possibly explained by the modulation of the ceramide system [17].”

P9, L205-212: “Inhibition of the ASM [37,52] by FIASMA medications may result in antiviral effects (through the diminution of ceramide-enriched membrane domains resulting in decreased viral entry and subsequent inflammation) and anti-inflammatory effects (through inhibition of this enzyme in endothelial and immune cells [9,11,12]). Because fluoxetine has the largest effect size in this study and one of the strongest in vitro effects on the ASM [52], is well-tolerated [53,54] and on the World Health Organization’s Model List of Essential Medicines, this molecule should be prioritized for randomized clinical trials in patients with COVID-19 [29].”

15/ Further details about why you hypothesized that FIASMA medication use would be associated with diminished mortality among COVID-19 inpatients.

Answer: This hypothesis was based on prior preclinical and clinical evidence. Following this comment, we have now provided more detail to support this point (please also refer to the response to comment #14):

P3, L94-106: “Several clinical trials have strengthened this preclinical evidence. Observational cohort studies of COVID-19 patients have indicated that FIASMA antidepressants and the FIASMAs amlodipine or hydroxyzine are associated with reduced risk of mechanical ventilation or death in the acute care setting [26–32], and decreased risk of hospital or emergency department visits in outpatients [33]. A systematic review and meta-analysis of 6 randomized controlled trials (RCTs) (N=4,197) found that medium-dose of the FIASMA antidepressant fluvoxamine (100 mg twice a day) was significantly associated with reduced mortality, hospitalization, and hospitalization/emergency department visits, and not associated with increased serious adverse events [34]. Finally, two observational multicenter retrospective cohort studies conducted at Greater Paris University Hospitals showed that FIASMA medications, mostly FIASMA antidepressants, calcium channel blocker medications, and hydroxyzine, were significantly associated with decreased likelihood of death or intubation [26,28], among inpatients with COVID-19.”

16/ Few sentences to describe your research design

Answer: Following this comment, we clarified the research design of the study in the abstract and in the method section.

Abstract: “We conducted a multicenter retrospective observational study including 72,105 adult patients with laboratory-confirmed SARS-CoV-2 infection who are admitted to 36 AP–HP (Assistance Publique–Hôpitaux de Paris) hospitals from May 2nd, 2020 to August 31st, 2022.”

P10, L248-255: “We conducted a multicenter retrospective cohort study at 36 AP-HP hospitals from May 2nd 2020 to August 31st 2022 [29], including all adults aged 18 years or over who had been hospitalized in these medical centers with COVID-19. COVID-19 was ascertained by a positive reverse-transcriptase–polymerase-chain-reaction (RT-PCR) test of nasopharyngeal or oropharyngeal swab specimens.”

P11, L298-307: “To examine the association between FIASMA medication use at baseline and hazards of mortality during the 28 days following admission, we performed Cox proportional hazards regression models [75] in a matched analytic sample of inpatients with COVID-19 receiving or not receiving a FIASMA medication. In order to reduce the effects of confounding, we used a 1:1 ratio matched analytic sample based on sex, age, hospital, period of hospitalization, medications prescribed as part of a clinical trial or according to compassionate use, psychiatric and other medical comorbidities, and biological and clinical markers of COVID-19 severity. Specifically, we used the nearest matching method [76]. We performed additional multivariable Cox regression models including all unbalanced covariate (i.e., with SMD>0.1) [73].”

P11, L311-319: “To test the robustness of the primary analysis, we performed sensitivity analyses and separately reproduced the above-mentioned analyses (i) in women and men, (ii) in younger and older patients (based on the median age of the full matched analytic sample), and (iii) in two different periods of hospitalizations (based on median date of hospitaliza-tion in the full matched analytic sample).

As an exploratory analysis, we reproduced the above-mentioned analyses for each class of FIASMA medications and individual FIASMA molecules. We selected a priori one control for each exposed case for exposures to each class of FIASMA medications, and 5 controls for each exposed case for exposures to each individual FIASMA molecule.”

We warmly thank the editors and reviewers for their help in improving our manuscript.

Reviewer 2 Report

The English language and grammatical mistakes can be improved. Do the necessary action before moving further with decision of this manuscript. 

Author Response

(The authors gave the same response as above.)

Reviewer 3 Report

Dear authors, 

thanks for your nice work, here are a few suggestions to improve the impact of your article:

1- in the introduction write more details about the mechanism of action behind the antidepressants or ambroxol effects

2- further details about why you hypothesized that FIASMA medication use 93 would be associated with diminished mortality among COVID-19 inpatients.

3- few sentences to describe your research design

minor editing of English language is required

Author Response

(The authors gave the same response as above.)

Round 2

Reviewer 2 Report

After careful evaluation of revised manuscript all the comments are properly addressed and the necessary corrections has been made. As per my recommendation I am satisfied with present form of manuscript and it is now suitable for publication in the PHARMACEUTICALS journal.